# Effects of Topical Hydrogen Purification on Skin Parameters and Acne Vulgaris in Adult Women

**DOI:** 10.3390/healthcare9020144

**Published:** 2021-02-01

**Authors:** Karolina Chilicka, Aleksandra M. Rogowska, Renata Szyguła

**Affiliations:** 1Department of Health Sciences, Institute of Health Sciences, University of Opole, 45-060 Opole, Poland; renata.szygula@uni.opole.pl; 2Department of Social Sciences, Institute of Psychology, University of Opole, 45-052 Opole, Poland; arogowska@uni.opole.pl

**Keywords:** hydrogen purification, acne vulgaris, alkaline water

## Abstract

Background: Acne vulgaris is a prevalent dermatological disease characterized by skin eruptions, which may decrease the sufferer’s quality of life. Hydrogen purification treatment is a new procedure used in cosmetology to improve the skin parameters of the face. This study examined the effectiveness of hydrogen purification treatment to improve women’s skin conditions with regard to acne vulgaris. Methods: In this study, 30 women participated who suffered from a high level of sebum and acne. The control group was comprised of 30 healthy women with a low level of sebum. The Hellgren–Vincent Scale and Derma Unit SSC 3 device were used to assess acne vulgaris severity and skin properties, respectively. Four hydrogen purification sessions were carried out at 7-day intervals, using the Hebe Hydrogenium+ generating alkaline water. Results: At baseline and 7 and 14 days after finishing the series of treatments, the levels of oiliness, moisture, and skin pH were tested. The main effects of treatment were significant in the following parameters: pH around the bottom lip, moisture between the eyebrows and around the nose, and oily skin in all three face sites. Conclusions: The level of sebum decreased and moisture levels increased during hydrogen purification. Topical hydrogen purification is an effective and safe treatment for acne vulgaris.

## 1. Introduction

The global prevalence of acne is 9.38% but varies widely due to the severity grading system, country, gender, and age, with rates being highest among adolescents in puberty (ranging from 35% to nearly 100%) [1]. Acne can persist in adulthood and can affect up to40% of all women [2]. A recent study showed that acne’s family history is a risk factor, but the severity of acne vulgaris (AV) is independent of premenstrual flares, diet, and body mass index (BMI) [3]. Problems with skin eruptions can negatively affect quality of life and lead to heightened stress, depression, or even suicidal ideations [4,5]. The effects of severe forms of acne are seen in deep acne scars, which can also decrease self-esteem and quality of life [4,5,6,7,8,9]. Thus, an interdisciplinary approach to treatment should be prioritized.

Among factors determining acne, the abnormal keratinization of the pilosebaceous canal, bacterial colonization (i.e., *Cutibacterium acne, Staphylococcus ureus,* and *Staphylococcus epidermidis*), increased production of sebum, skin inflammatory, genotypic factors, and hormonal disorders can be distinguished. Treatment is often long and tedious, affecting the patients’ sense of hopelessness and social isolation. In recent years, considerable attention has been paid to drug use as the best treatment for acne. Psychological support is currently also recommended to improve acne [10,11,12,13,14]. However, new cosmetology acne treatments, such as acid peels, microdermabrasion, cavitation peeling, sonophoresis, and hydrogen purification treatment, are continually being sought.

Hydrogen has been known in the world of science for decades. It is the simplest molecule in nature, created by two hydrogen atoms. Ohsawa et al. [15] showed that molecular hydrogen could selectively reduce reactive oxygen forms in vitro and exert antioxidant activity. Thanks to this discovery, hydrogen gas has been used in 300 diseases and have come to the forefront of therapeutic medical gas research. Clinical research has shown that hydrogen is an essential physiological regulator factor with anti-inflammatory and antioxidant effects on cells and organs [15,16,17,18,19].

Hydrogen purification treatment uses alkaline water (electrochemically reduced water, ERP), which is rare in nature. However, with the chemical process of electrolysis, ERP can be produced immediately with a pH range between 8–10. It has both high alkaline and negative oxidation-reduction potential (ORP). The electrolysis process consists of passing a direct current between two electrodes (anode and cathode) separated by a semi-permeable membrane. During this process, the elements contained in the water break down into hydrogen ions H+, focusing on the cathode, and hydroxy ions OH-, which focus on the anode. Ionized alkaline water consists of H+ ions; it also contains alkaline elements, such as calcium, magnesium, potassium, hydrogen, and alkaline hydroxyl ions. Negative OH-ions form acid water, containing chlorine, sulfur, phosphorus, positive acid ions, and oxygen.

The main factor responsible for the ORP of alkaline water is the active molecular hydrogen in alkaline water. Its concentration, depending on the concentration of alkaline water, ranges from 0.3 to 0.6 mg/L. Only the electrolysis processes (water ionizers) and hydrogenation (hydrogen generators) allow water production with active hydrogen ions, which is a natural and potent antioxidant. Currently, hydrogen cleansing is commonly used in cosmeceutical treatments to reduce free radicals and achieve anti-aging properties. Reactive oxygen species bind to hydrogen ions, forming water and pure oxygen. Through chemical processes, hydrogen purification neutralizes free radicals, and hydrogen water significantly reduces lubrication on the skin’s surface.

In recent years, research has been conducted into the internal effects of alkaline water on patients affected by such problems as pyrosis, dysphoria, abdominal distension, and chronic diarrhea [20,21,22,23,24]. The research confirmed that the drinking of alkaline water for two weeks resulted in a significant health status improvement. However, little is known about alkaline water’s superficial effect on selected skin parameters in people suffering from acne vulgaris. To the best of our knowledge, the present study explores, for the first time, the effectiveness of alkaline water in the treatment of acne vulgaris.

## 2. Materials and Methods 

### 2.1. Participants in the Study

This prospective case-control clinical study with follow-up analysis was conducted between January 2020 and February 2020 at Opole Medical School in Poland. It was approved by the Human Research Ethics Committee of the Opole Medical School (KB/54/NOZ/2019), according to the principles of the Declaration of Helsinki. The study was registered at the International Standard Randomized Controlled Trial Number (ISRCTN) registry database (No. ISRCTN 13842359). All participants signed written consent before the beginning of the study regarding participation in the study and the use of photographic images for publication. A photo was taken before treatment and 14 days after finishing all four treatment sessions (front view, left oblique, and right oblique). The subjects were informed that they could withdraw from the study at any time without giving a reason. Initially, 123 people participated in the study, but 35 individuals met the exclusion criteria, and another eight persons did not meet the inclusion criteria. After excluding 43 people from the study, the final study sample consisted of 80 women aged between 20 and 23 (*M* = 21.23; *SD* = 0.56).

Group 1 included 40 young adult women with a higher sebum level and acne vulgaris (AV group). Participants were diagnosed with mild facial acne vulgaris using the Hellgren–Vincent Scale (HVS) to determine acne lesions’ severity. The HVS can accurately determine the number of skin lesions in the form of lumps, pustules, and blackheads. The AV group’s inclusion criteria were a young age, ranging between 18–25 years old, and severe acne, with a second degree in the HVS (*n* = 23, 76.67%) and a third-degree in the HVS (*n* = 7, 23.33%). The exclusion criteria for the AV group were severe acne, antibiotics, and retinoids use for acne treatment within the previous four weeks, medication use that could aggravate or suppress acne (vitamin B, halogens, antiepileptics, antidepressants, ciclosporin), oral contraceptive pill, oral antibiotics use within the previous three months, pregnancy, breastfeeding, active inflammation of the skin, bacterial, viral, allergic and fungal relapsing skin diseases, disturbed skin condition, recent surgical procedures in the treating areas, active reduced immunity, epilepsy, claustrophobia, active rosacea, eczema, psoriasis, numerous telangiectasias, numerous melanocytic nevi, tanned skin, skin cancers, and recently having surgery (up to 2 months). Table 1 demonstrates the prevalence of acne severity in AV group.

Group 2—the control sample (CS)—consisted of 40 healthy young adult women without any previously diagnosed dermatological problems or dermatological disease. Individual interviews assessed these issues. The inclusion criteria for the CS were a young age, ranging between 18–25 years old, good overall health condition, and lack of dermatological problems. The exclusion criteria for the CS group included pregnancy, breastfeeding, active inflammation of the skin, bacterial, viral, allergic, and fungal relapsing skin diseases, disturbed skin condition, new surgical procedures in the treating areas, active reduced immunity, epilepsy, claustrophobia, active rosacea, eczema, psoriasis, numerous telangiectasias, numerous melanocytic nevi, tanned skin, skin cancers, and recently having surgery (up to 2 months).

No other treatments were allowed during the study for both groups. During the treatment series, 10people dropped out of both the AV and CS groups. Some of the people did not appear at subsequent treatments. Some participants from the CS reported excessive dryness of the skin after performing hydrogen cleansing. Finally, 30 women from the AV group and 30 individuals from CS completed the study. The consort flow chart of the clinical study is presented in Figure 1.

### 2.2. Apparatus for Hydrogen Purification Treatment

Hebe Hydrogenium+ was used for a series of hydrogen purification treatments (Figure 2). The Hydrogenium+ is an innovative device for non-invasive and comprehensive skincare. Patients from both AV and CS groups underwent four sessions of hydrogen purification treatment using the Hydrogenium+ device. The device has two cups: Chalice 1 is used for the production of alkaline water and contains a generator with intake pipes, which allows the relentless determination of the part the alkaline water is in (with the addition of calcium, magnesium, and potassium ions); Chalice 2 is intended for used water.

Chalice 1 is equipped with a water-insoluble membrane that stabilizes the system (nanosilver) and two electrodes made of titanium, covered with a 25-micron layer of platinum (thanks to which the electrolysis process takes place). When low-mineral water (below 500 mg/L) is poured into Chalice 1, the device is switched on with the start button. It is ready when it shows 100% preparation on the panel. In Chalice 1, on one side of a membrane, alkaline water is located, which is used for the treatment, while on the second side of the membrane, acidic water is placed (with chlorine and sulfur ions). The device draws water in an alkaline environment. The parameters of the electric current used during the treatment are 1 ampere and 150 volts. The water temperature is 25 °C when heated by the device.

### 2.3. Treatment Procedure

A cosmetic makeup removal (micellar fluid) was performed at the first step of the procedure. Next, hydrogen cleansing was conducted using the H2 peel head, sucked the skin fold under a vacuum (supplying clean water with one cable), and flushed it with hydrogenated water. Used water was sucked in the second wire connected to the head, which drained it to the wastewater tank. The device’s power was set for each of the four treatments at about a 10% vacuum; a higher dose could cause the appearance of ecchymosis on the skin. This stage of the procedure was performed for 10 min.

The next stage of the procedure was applying an H2 jet head, using the potential of active hydrogen (hydrogenated water injection), which had a nozzle “ejecting” the hydrogenated water under pressure. Power of 2 bars (intended for the face area) was used here for each series of treatments. This head was guided at a distance of 2–3 cm from the client’s skin for 5 min. Participants were treated at a 1-week interval for four sessions.

After the sessions, only moisturizing cream was applied to the facial skin of the subjects. Home care was recommended, which consisted of washing the face with micellar fluid and applying only a moisturizing cream. During the study, the participants were instructed that it was forbidden to perform any other cosmeceutical, exfoliating, or apparatus procedures.

### 2.4. Skin Parameter Assessment

The Hellgren–Vincent Scale (HVS) was used for the clinical evaluations of acne vulgaris severity. The HVS is a five-point scale in which a higher degree means a higher level of acne, as follows: (1) erythema, blackheads, 1–5 papules or pustules; (2) erythema, blackheads, 6–10 papules or pustules; (3) erythema, blackheads, 11–20 papules or pustules; (4) erythema, blackheads, 21–30 papules or pustules; (5) erythema, blackheads, more than 30 papules or pustules [25]. The HVS was assessed before the treatment and after 14 days from finishing four sessions of hydrogen purification. An example of a participant from AV and SC samples is shown in Figure 3.

The skin parameters’ functional measurements were performed twice (before treatment, and 7 and 14 days after the complete session of four treatments), using a Derma Unit SSC 3 device (Courage + Khazaka Electronic GmbH, Cologne, Germany). The following skin parameters were measured: sebum (using Sebumeter, Courage + Khazaka Electronic GmbH, Cologne, Germany), skin surface pH (using a Skin-pH-Meter, Courage + Khazaka Electronic GmbH, Cologne, Germany), and skin moisture (using the Corneometer, Courage + Khazaka Electronic GmbH, Cologne, Germany). Three measurements were taken in the face of each participant: (1) between the eyebrows; (2) about 1 cm from the lobe of the nose (left and right); and (3) about 1 cm from the lower lip (on the chin)—see the Appendix A. Photos were also used to assess acne lesions with the hydrogen purification treatment. The clinical photographs were taken at baseline and 14 days after finishing four sessions of hydrogen purification. 

### 2.5. Statistical Analysis

Firstly, acne vulgaris severity changes were assessed by comparing the HVS at baseline and 14 days after finishing four sessions of hydrogen purification in the AV group. The non-parametric Wilcoxon matched-pairs test was used for this purpose. Secondly, descriptive statistics were calculated separately in AV and control groups for each skin parameter, including the number of observations, mean, standard deviation, minimum, and maximum. The normality of the distribution of the data sets was examined using the Kolmogorov–Smirnov *d* test. 

Because all data showed normal distribution in both groups (*p* > 0.05), the ANOVA with repeated measures (at baseline and on weeks 1 and 2 after finishing treatment of hydrogen purification) was performed to compare skin parameters (such as pH, sebum, and moisture) in selected places of the face (between the eyebrows, around the nose and around the bottom lip) in both groups (AV and control). Fisher’s least significant difference (LSD) post-hoc test was conducted to find significant differences between groups, as well as between treatment conditions: test (at baseline) and retest (7 and 14 days after finishing treatment). Also, the effect size for ANOVA tests was calculated using a partial eta-square coefficient. All statistics were calculated using Statistica 13.1. Software (Round Rock, TX, U.S.).

## 3. Results

The number and percentage of women with a particular degree of acne vulgaris on the Hellgren–Vincent Scale are presented in Table 1. At baseline, most participants in the AV group were diagnosed with a second degree (76.67% of the sample) and several people with a third-degree (23.33%) of AV. Two weeks after finishing hydrogen purification treatment, around half of the participants were diagnosed with the first (53.33%) and second (46%) AV degree. This shows that the level of AV decreases under the influence of hydrogen purification treatment in most individuals, which was confirmed by the Wilcoxon matched-pairs test, *n* = 21, T = 0.00, Z = 4.02, *p* < 0.0001. Changes in the HVS due to hydrogen purification treatment are shown in Figure 3 and Figure 4.

Comparison of skin parameters between AV and control groups is shown in Table 2. Statistically significant group differences were found for sebum and moisture at all three measuring sites of the face (between the eyebrows, around the nose, and the bottom lip). The AV groups demonstrated higher scores for oiliness and lower scores for moisture than the control group.

The treatment was significant in the following parameters: pH around the bottom lip, moisture between the eyebrows and around the nose, and oily skin in all three face sites. According to expectations, the sebum level decreased and moisture levels increased under the influence of hydrogen purification treatment. Fisher’s LSD post-hoc test showed significant differences between the baseline test and retest after the first session (*p* < 0.0001). The differences between the first measurement at baseline and two weeks after the end of all four treatment sessions persisted (*p* < 0.0001). However, the second and third measurements (7 and 14 days after the last sessions) did not differ significantly from each other (*p* > 0.05). An interaction effect between groups and treatments was also found for oily skin in all three sites on the face. The Fisher’s LSD post-hoc test indicated that treatment was effective in the AV group (*p* < 0.0001), but the oily skin among CS participants did not change under the hydrogen purification treatment (*p* > 0.05). More details are presented in Figure 5, Figure 6 and Figure 7.

## 4. Discussion

This 4-week study demonstrated that the hydrogen purification treatment designed specifically for women with mild to moderate acne led to a significant improvement in the severity of acne and the overall appearance and condition of the skin. To the best of our knowledge, a hydrogen purification treatment was examined regarding the topical cosmeceutical treatment of acne vulgaris for the first time in this study. The current treatment is based on active hydrogen. As one of the most powerful antioxidants, active hydrogen counteracts the aging process and alleviates inflammation and oxidative stress. The high-quality Hydrogenium+ generator removes oxygen and unfavorable ions of chlorine, sulfur, and phosphorus from the water, saturating it with active hydrogen and beneficial calcium, magnesium and potassium ions. Hydrogen can easily penetrate deep into the skin structures, bound with the most harmful free oxygen radicals, which effectively neutralizes them and deprives the skin of harmful properties.

Then, as water, it is excreted from the body. Hydrogen purification can improve the skin’s efficient functioning, slowing down the aging process, and reducing the effects of stress and environmental pollution. 

This study demonstrated that hydrogen’s antioxidant activity mightalso affect common skin pathogens (e.g., *Cutibacterium acne, Staphylococcus ureus, Staphylococcus epidermidis*). It seems possible that the hydrogen may alleviate acne due to its antimicrobial properties and by reducing the sebum level. Sebum plays a key role in the pathogenesis of acne [26,27,28]. In turn, the reduced sebum levels can improve skin properties and reduce acne.The inflammation’s progress in the pathogenesis of acne may be related to sebum changes and reactive oxygen species (ROS) releasing from the impacted damaged follicular walls. Some of the common drugs in acne treatments, such as hydrogen peroxide, function by reducing ROS [29]. A previous study showed that adding hydrogen peroxide to the preoperative skin preparation may be a low-risk method to reduce deep tissue contamination with *C*. acnes [30,31,32]. Moreover, acne patients are recommended alkaline soaps (pH 9–10) with various surfactants [33,34]. Hydrogen sulfide concentrations in thermal spring waters showed inhibition properties on human origin’s bacterial proliferation [35].

This study indicated that hydrogen purification treatment might positively reduce skin eruptions and increase moisture, as well as decrease the skin’s oiliness in people suffering from acne vulgaris. In particular, the sebum level was significantly decreased only in the AV group, while skin moisture between the eyebrows and around the nose was significantly increased in both the AV and CS groups. The appropriate level of skin hydration plays a protective role in the skin barrier. Alkaline water may moisturize the skin and improve the skin’s protective barrier. However, optimal times and methods of daily cleansing and moisturizing facial skin may have an equally beneficial effect on healthy skin [36]. Many cleansers also moisturize, prevent, and soothe skin irritation by slowing down the evaporation of water.

The positive effect of the hydrogen purification treatment on the reduction of sebum is probably due to alkaline water’s beneficial effects. Alkaline water has a high level of pH, which can reduce excess sebum from the epidermis. Alkaline water seems to play a vital role as a regulatory factor with anti-inflammatory and antioxidant effects on cells and organs [15]. Some skin cleansers, such as soap (pH 9–10), have a similar pH to alkaline water. However, washing a face with such a daily preparation would irritate the skin. The hydrogen purification treatment was performed on our patients once a week, thanks to which it brought excellent results in terms of reducing sebum without any side effects [37].

The hydrogen purification treatment did not significantly change the pH of the participant’s facial skin in both AV and CS samples. This result is consistent with previous studies, which indicated that none of the five facial areas (i.e., forehead, nose, chin, and right/left cheeks) differed significantly in pH between healthy participants and acne individuals [38]. The pH of the skin surface has an impact on the barrier function [34]. The current results showed that the acidification of the examined people’s skin was in the range of 4.1–5.8 pH, which is the norm for the general population [39]. Schürer [40] found that a low pH level has skin protective properties, slowing down bacteria’s growth. On the other hand, a large increase in pH may damage the skin surface, causing irritation, which may adversely affect the skin microbiome. Various external and internal factors can affect skin pH [34]. Skin moisture, sweat, sebum, anatomy, genetic predisposition, age, and diet are among the many endogenous factors influencing skin pH. Exogenous factors include detergents, application of cosmetic products, occlusive dressings, topical antibiotics, humidity, season, and individual differences in behavioral habits. The current study found that hydrogen purification did not affect facial skin pH in young women. Perhaps this treatment was too short to change the pH significantly. Further research is necessary to explain this question.

Previous studies only examined the healing properties of alkaline water for internal use by drinking it. For instance, Gadek et al. [41] found that the daily drinking of 2 L of alkaline water (Nordenau and Hita Tenryosui water) decreased glucose HbA1c levels in people with type 2 diabetes. Significant changes were observed after just six days, and prolonged use of alkaline water caused a decrease in blood cholesterol and creatinine, as well as an increase in density lipoprotein levels [41,42]. Furthermore, Osada et al. [43] showed that drinking 2 L alkaline water (Hita Tenryosui water) per day for two months significantly decreased blood sugar levels in 89% of diabetes patients. Similarly, blood triglyceride and total cholesterol levels significantly decreased in 92% of patients with hyperlipidemia. The study conducted by Higashikawa et al. [44] showed that drinking alkaline water has an anti-metabolic effect, demonstrated by decreased starved blood sugar levels, blood pressure, and total cholesterol LDL cholesterol, GOT, g-GTP, triglyceride levels, arteriosclerosis index, and uric acid levels.

In contrast, an increase was noted in leptin levels, as well as the improvement of constipation. Alkaline water has also been used to support cancer treatment [45,46,47,48]. Alkaline water suppresses tumor angiogenesis, inhibits scavenging intracellular ROS, suppresses gene expression and vascular endothelial growth factor secretion, inhibits cancer cells and microorganisms’ growth, and induces apoptosis along with glutathione in HL60 human leukemia cells. The ease of formation of alkaline water and its universality can bring about many health-related effects in the treatment of many disorders, including dermatological diseases and the maintenance of health. Thus, the promising results of this study should be continued and confirmed in the future.

Previous research has shown the effectiveness of cosmeceutical treatment for improving skin parameters and reducing the scars of people with acne vulgaris, using acid peels, radiofrequency, and fractional microneedle radiofrequency, microdermabrasion, sonophoresis, and intense pulsed light (IPL) [25,49,50]. Compared to other topical cosmeceutical treatments, such as cosmetic acids [9,50], the hydrogen purification treatment is not associated with the risk of irritation, reddening of the skin, or side effects as burning and itching of the epidermis surface. People with sensitive skin, prone to irritation and with certain undesirable symptoms related to the type of skin (e.g., dryness, tightness, coarseness, washout of makeup, quick production of sebum, feeling differences in oiliness, frequencies of acne lesions, and oiliness) [51], should pay special attention to treatments with the use of cosmetic acids or other topical therapies [52]. However, hydrogen purification appears safe and effective in improving this type of skin [53].

It should be added that hydrogen purification is not a treatment and cannot displace dermatological methods [52,54]. However, it can support the other dermatological drug and topical therapies [27] in moisturizing and reduce the level of sebum and skin lesions among people with mild and moderate acne. For people with fistulas or cysts, the procedure is prohibited, and solely dermatological treatment is possible. It should be emphasized that the growing resistance to antibiotics in *Cutibacterium* acnes and other bacteria is becoming extremely alarming. Hydrogen purification can be used as an auxiliary in people who have developed drug resistance due to excessive long-time use of preparations medicinally, and also in those individuals who cannot use topical agents in the form of creams or ointments due to numerous side effects, such as severe irritation or peeling of the skin [55,56].

### Study Limitation

The present study used the HVS to assess the severity of acne vulgaris and the Derma Unit SSC 3 device to measure selected skin parameters, such as the skin’s pH, oiling, and moisture. In the future, we would like to use more measuring tools; for example, a specialist camera could improve the test quality of the study. The Visiopor^®^ PP 34 (Courage + Khazaka Electronic GmbH, Cologne, Germany) camera uses a specific UV-light to visualize the fluorescing acne lesions of an area of at minimum 8 × 6.4 mm. The orange-red fluorescence indicates Propionibacterium acne bacteria’s presence within clinically non-evident (follicular impactions and micro comedones) and clinically evident (comedones, papules, and pustules) lesions.

The study’s other limitation is the research sample, i.e., the study focused on a homogeneous group of young women. Moreover, the sample size in this study was not large. In the future, a larger sample size should participate in the study, including both women and men with acne in a broader range of age and people without any skin problems or disease (for comparison). Further, it would be valuable to compare the effectiveness of the hydrogen purification treatment using alkaline water with other cosmetic treatments using cosmetic acidic pH substances and other topical cosmetics treatments. 

## 5. Conclusions

Hydrogen purification is an effective and safe treatment for acne vulgaris. In this work, researchers conducted a study on people struggling with acne vulgaris and showed that hydrogen purification could help treat this disease for the first time. Treatment with hydrogen purification is safe and effective in reducing acne skin eruptions and can improve the skin’s condition and an appearance by decreasing sebum and increasing the moisture level in young women. In combination with oral drugs and other topical therapies, hydrogen purification may be considered to optimize acne treatment outcomes and overcome the problems encountered with some conventional methods. However, it is essential to note that hydrogen purification cannot replace drug treatments but may be used as a supportive topical cure for acne vulgaris, combined with other forms of therapy, according to clinicians’ recommendations [57].

## Figures and Tables

**Figure 1 healthcare-09-00144-f001:**
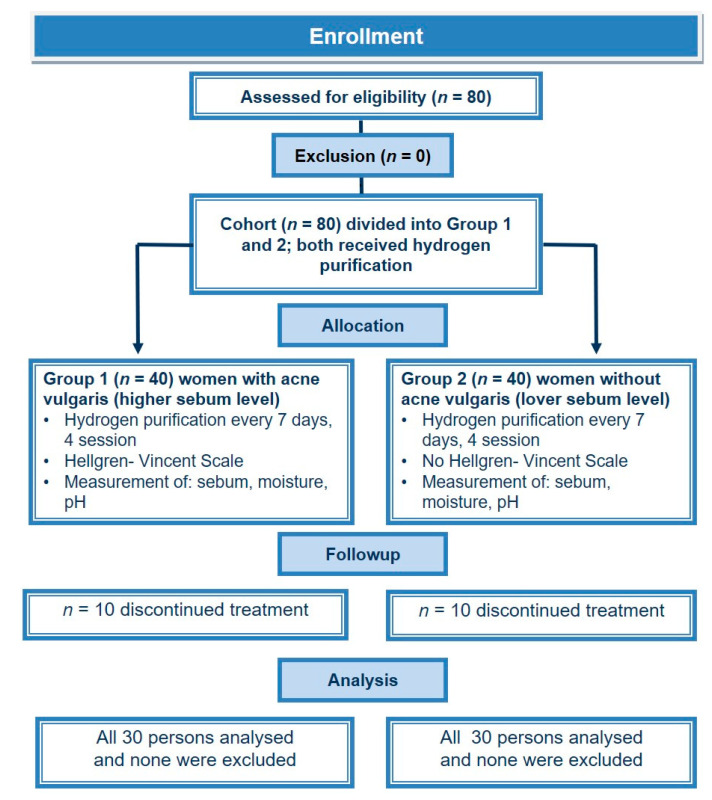
Consort flow chart of clinical study group 1 (acne vulgaris, AV) and group 2 (control sample, CS).

**Figure 2 healthcare-09-00144-f002:**
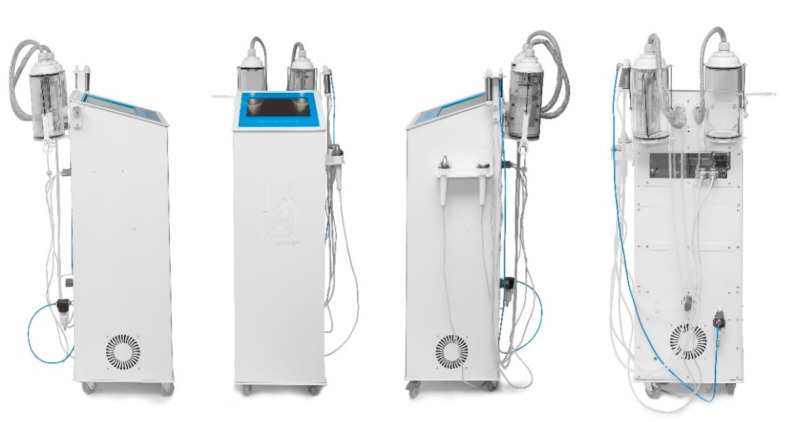
Hebe Hydrogenium+.

**Figure 3 healthcare-09-00144-f003:**
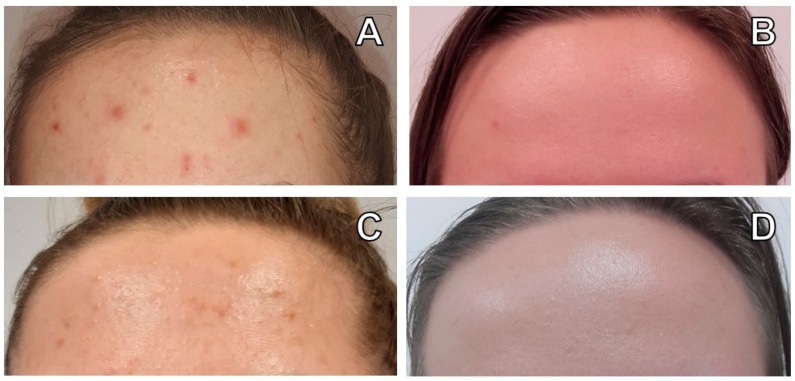
Example of participant’s forehead: (**A**) from AV group before hydrogen purification (HP); (**B**) from CS group before HP; (**C**) from AV group 14 days after HP; (**D**) from CS group after HP.

**Figure 4 healthcare-09-00144-f004:**
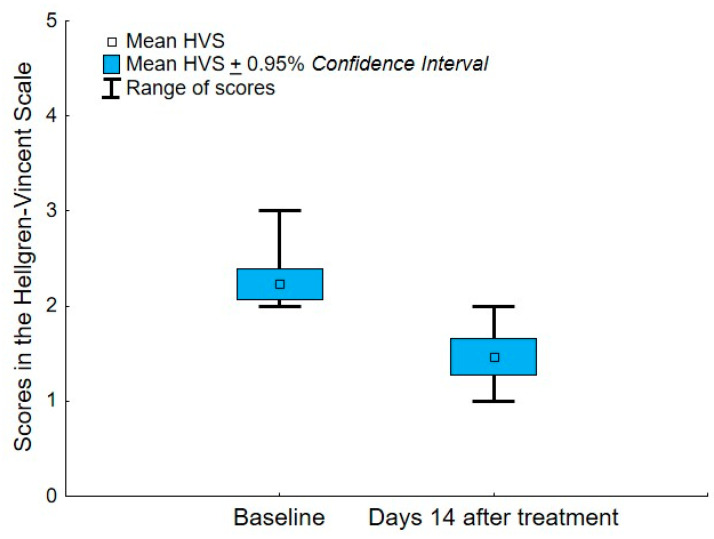
Differences in the HVS between two measurements: at baseline and 14 days after finishing the 4-week treatment of hydrogen purification.

**Figure 5 healthcare-09-00144-f005:**
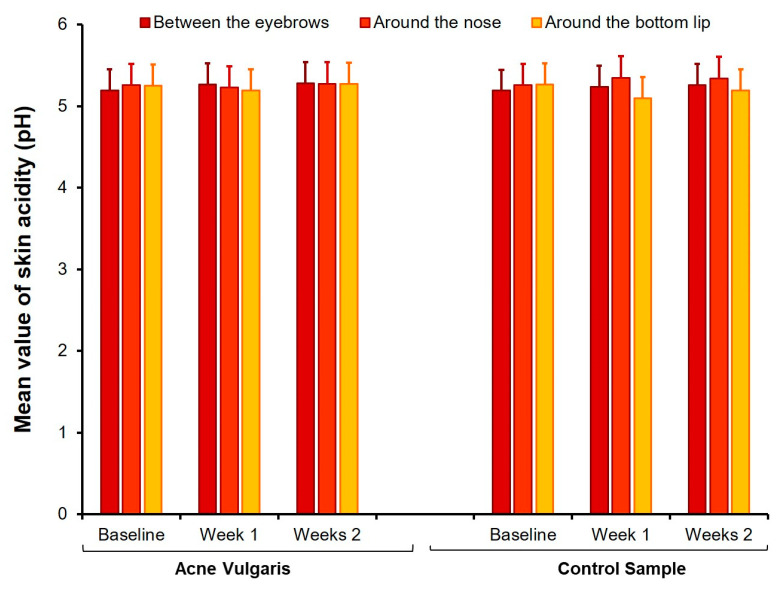
Comparison of mean values of skin acidity at baseline, one week, and two weeks after finishing the treatment with hydrogen purification in both AV and CS groups. Error bars are standard deviations.

**Figure 6 healthcare-09-00144-f006:**
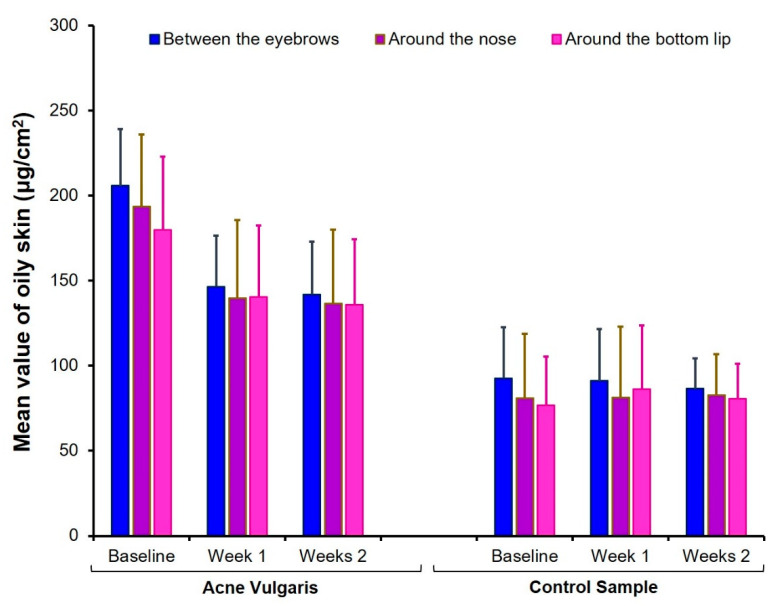
Comparison of mean values of oily skin at baseline, one week, and two weeks after finishing the treatment with hydrogen purification in both AV and CS groups. Error bars are standard deviations.

**Figure 7 healthcare-09-00144-f007:**
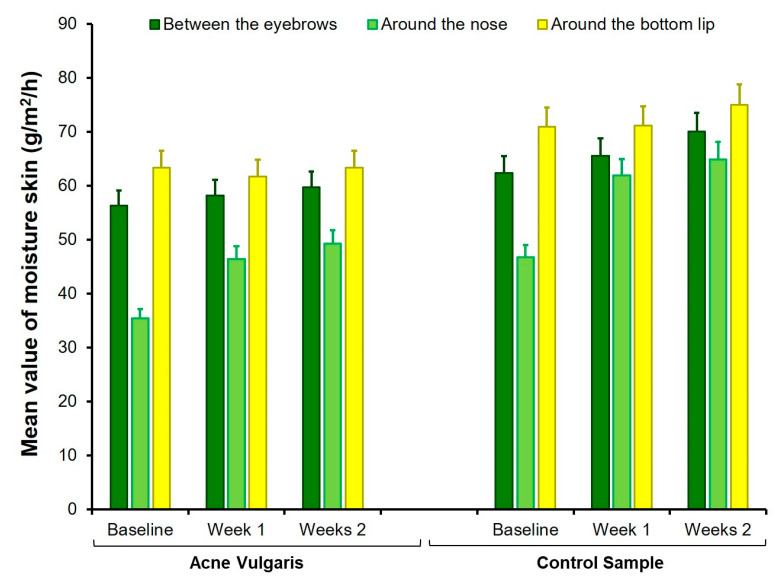
Comparison of mean values of moisture skin at baseline, one week, and two weeks after finishing the treatment with hydrogen purification in both AV and CS groups. Error bars are standard deviations.

**Table 1 healthcare-09-00144-t001:** Characteristics of acne vulgaris severity in terms of the Hellgren–Vincent Scale (HVS) among participants of the acne vulgaris (AV) group (*n* = 30).

	Baseline		Days 14 after Finishing Treatment	
HVS	*n*	%	*n*	%
I	0	0.00	16	53.33
II	23	76.67	14	46.67
III	7	23.33	0	0.00

**Table 2 healthcare-09-00144-t002:** Means, standard deviations, and two-way ANOVA with repeated measure statistics for study variables.

	AV (*n* = 30)		CS (*n* = 30)		ANOVA			
Variable	*M*	*SD*	*M*	*SD*	Effect	*F ratio*	*df*	η_p_^2^
PH between the eyebrows								
Baseline	5.19	0.45	5.19	0.52	G	0.05	1, 58	0.00
Week 1 after treatment	5.27	0.40	5.24	0.42	T	1.45	2, 116	0.02
Week 2after treatment	5.28	0.26	5.26	0.18	G × T	0.04	2, 116	0.00
PH around the nose								
Baseline	5.26	0.47	5.26	0.47	G	0.61	1, 58	0.01
Week 1 after treatment	5.23	0.29	5.35	0.35	T	0.68	2, 116	0.01
Week 2 after treatment	5.28	0.25	5.34	0.28	G × T	0.98	2, 116	0.02
PH around the bottom lip								
Baseline	5.25	0.40	5.27	0.47	G	0.52	1, 58	0.00
Week 1 after treatment	5.20	0.29	5.10	0.31	T	4.19 *	2, 116	0.07
Week 2 after treatment	5.27	0.22	5.19	0.27	G × T	1.18	2, 116	0.02
Oiliness between the eyebrows								
Baseline	205.87	33.23	92.73	29.85	G	147.99 ***	1, 58	0.72
Week 1 after treatment	146.60	30.05	91.07	30.45	T	50.93 ***	2, 116	0.47
Week 2 after treatment	141.90	31.02	86.53	17.79	G × T	38.86 ***	2, 116	0.40
Oiliness around the nose								
Baseline	193.77	42.12	80.97	37.90	G	68.29 ***	1, 58	0.54
Week 1 after treatment	139.67	45.87	81.30	41.65	T	28.57 ***	2, 116	0.33
Week 2 after treatment	136.63	43.27	82.77	24.06	G × T	30.97 ***	2, 116	0.35
Oiliness around the bottom lip								
Baseline	179.87	43.22	76.63	28.73	G	83.17 ***	1, 58	0.59
Week 1 after treatment	140.40	42.18	86.40	37.28	T	11.07 ***	2, 116	0.16
Week 2 after treatment	135.87	38.38	80.63	20.67	G × T	20.25 ***	2, 116	0.26
Moisture between the eyebrows								
Baseline	56.37	17.30	62.37	15.07	G	4.20 *	1, 58	0.07
Week 1 after treatment	58.21	16.90	65.59	16.31	T	4.78 **	2, 116	0.08
Week 2 after treatment	59.71	19.08	70.06	16.78	G x T	0.78	2, 116	0.01
Moisture around the nose								
Baseline	35.40	17.76	46.75	23.34	G	10.92 **	1, 58	0.16
Week 1 after treatment	46.49	20.85	61.92	17.87	T	24.74 ***	2, 116	0.30
Week 2 after treatment	49.28	19.91	64.89	18.45	G × T	0.50	2, 116	0.00
Moisture around the bottom lip								
Baseline	63.33	13.71	71.00	18.83	G	5.15 *	1, 58	0.08
Week 1 after treatment	61.77	16.15	71.19	18.88	T	1.79	2, 116	0.03
Week2 after treatment	63.36	16.33	75.02	21.09	G × T	0.91	2, 116	0.02

Note: N = 60. ANOVA = analysis of variance; AV = group with acne vulgaris (*n* = 30); CS = control group without acne vulgaris; G = group; T = treatment of hydrogen purification. Oily skin parameter is expressed in units μg/cm^2^; moisture in g/m^2^/h.* *p* < 0.05, ** *p* < 0.01; *** *p* < 0.001.

## Data Availability

The data presented in this study are available in the Appendix A.

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
