# Peer review of "Effects of Topical Hydrogen Purification on Skin Parameters and Acne Vulgaris in Adult Women"

_healthcare, 2021, doi:10.3390/healthcare9020144_

Round 1
Reviewer 1 Report
This work by Chilicka et al.; describes the effect of Hydrogen purification towards improving acne vulgaris in female patients.
The study mades use of a commercial medical device to perform the hydrogen treatment and evaluated its effects on acne by using the Hellgren–Vincent Scale, skin ph, moisture and oily skin. Their results indicate benefits for the use of hydrogen purification in the treatment of acne. Altough the results seem promising, the authors provide limited insight into their experimental results and do not provide much justification on how hydrogen purification will complement or replace current treatments. In general, the study is well structured and provides an interesting potential dermal application. Thus I would accept the publication of this work after some minor concerns are addressed.
1 Can the authors expand the implication of their results? For instance, the ph and oily data are presented but not discussed in detail, or insight is offered.
2 Can the author justify when hydrogen purification is warranted or preferred1 compared to the gold standard?
3 I will suggest also include graphs for ph and moisture as done with oily skin and HVS.
4 If possible, the inclusion of actual photographs of patients before and after treatment could serve as a more impactful proof of the potential of this treatment.
5 As this is a novel treatment, a scheme or actual picture of the medical device used for the hydrogen purification treatment would help introduce it to a more general audience.
Reviewer 2 Report
This is a well presented article of the potential use of the effect of hydrogen purification in acne vulgaris in young females. However, I have some constraints regarding the methodology used:
- At your introduction you mention that 80-100% of people between the age of 11 and 30 have acne vulgaris, where did you get these numbers? The overall prevalence of AV is estimated around 9%, while even in adolescents the range is way wider with the range starting at around 35% (https://www.nature.com/articles/s41598-020-62715-3#:~:text=Acne%20Epidemiology,ages)%20of%209.38%251.)
- I am quite surprised that none of the 80 patients assessed for eligibility met any of your exclusion criteria. For example, use of oral contraceptives by young females in Poland was recently found to be at 38%, yet none of your young individuals was on OCs? (https://www.ncbi.nlm.nih.gov/pmc/articles/PMC6695758/#!po=18.7500)
- Additionally, as you mention in your limitations, the patient sample studied is quite biased, yet balanced for your control group
Round 2
Reviewer 2 Report
I think that the manuscript has been improved and is now suitable for publication.
This manuscript is a resubmission of an earlier submission. The following is a list of the peer review reports and author responses from that submission.